# Judgement Bias in Miniature Donkeys: Conditioning Factors and Personality Links

**DOI:** 10.3390/ani11092737

**Published:** 2021-09-19

**Authors:** Maria Pinto, Francisco Javier Navas González, Camie Heleski, Amy McLean

**Affiliations:** 1IFM Biology, Linköping University, 58183 Linköping, Sweden; mpalmapinto@gmail.com; 2Institute of Agricultural Research and Training (IFAPA), Alameda del Obispo, 14004 Córdoba, Spain; 3Department of Genetics, Faculty of Veterinary Sciences, University of Córdoba, 14071 Córdoba, Spain; 4The Worldwide Donkey Breeds Project, Faculty of Veterinary Sciences, University of Córdoba, 14071 Córdoba, Spain; acmclean@ucdavis.edu; 5College of Agriculture, Food and Environment, University of Kentucky, Lexington, KY 40546, USA; camie.heleski@uky.edu; 6Department of Animal Science, University of California, Davis, CA 95616, USA

**Keywords:** cognition, optimism, pessimism, equid, patience

## Abstract

**Simple Summary:**

Optimism and pessimism may affect the way individuals perceive elements of the environment which they are surrounded by, but the mechanisms behind these processes are yet to be thoroughly described. The present study addresses judgement bias and its correlation with personality in Miniature Donkeys. Individuals were scored on eighteen personality traits and their response to an ambiguous stimulus. Judgement bias presents intrinsic individual differences. The correlation found between patience and pessimism suggests that personality-related conditions may shape the way individuals interpret new stimuli. Improving our knowledge of tools that measure donkeys’ mood may play a pivotal role from an animal welfare perspective, as it may provide a better understanding of individuals’ interaction among handlers, congeners, and with the environment, and issue their own verdicts after such interactions.

**Abstract:**

Expectation-related bias may configure individuals’ perception of their surrounding environment and of the elements present in it. This study aimed to determine the repercussions of environmental (weather elements) or subject-inherent factors (sex, age, or personality features) on judgment bias. A cognitive bias test was performed in eight Miniature jennies and four jacks. Test comprised habituation, training and testing phases during which subjects were trained on how to complete the test and scored based on their latency to approach an ambiguous stimulus. A questionnaire evaluating eleven personality features was parallelly completed by three caretakers, five operators and two care assistants to determine the links between personality features and judgment bias. Adjusted latencies did not significantly differ between sexes (Mann–Whitney test, *p* > 0.05). Although Miniature donkeys can discriminate positive/negative stimuli, inter-individual variability evidences were found. Such discrimination is evidenced by significant latency differences to approach positive/negative stimuli (33.7 ± 43.1 vs. 145.5 ± 53.1 s) (Mann–Whitney test, *p* < 0.05). Latencies significantly increased with patience, indicative of an expression of pessimism. Better understanding judgement bias mechanisms and implications may help optimize routine handling practices in the framework of animal welfare.

## 1. Introduction

A cognitive bias is an adaptation to the environment with which an individual interacts and which translates into a systematic pattern that deviates from norm or logic in judgment [1]. Pessimism or optimism are specific examples of cognitive biases which serve to balance the outcomes, either negative or positive, that an individual could expect out of an interaction with their environment. Contextually, judgement bias tests evaluate the degree to which individuals deviate from the norm, how such biases lead to the increase or reduction in the probability to receive a positive/negative outcome, and how these satisfactory or unsatisfactory experiences remain or condition future individual–environment interactions [2]. Human and nonhuman individuals may accommodate their actions to seek the fulfilment of their expectations in the accomplishment of particular aims. The retrieval of past experiences, current life conditions, personality feature expressions, state of mind [3], or anecdotal events can turn into judgement patterns for particular situations, which sometimes leads to distorted perception and illogical interpretation of events [3].

Cognitive bias has also been identified as state or affect-modulated [4] or induced [5] cognition, hence, it has been used describe the influence of affective state on information processing in nonhuman animals [5]. In this context, a state can be defined within the scale of how much an individual has to do in order to cope with the environment, and how well coping attempts succeed [5]. In view of this definition, a negative state would result after an individual makes a bigger effort or an increased number of repeated efforts than those it intended or was able to make at a particular moment when coping with a specific situation.

Paul and Mendl [6] defined emotion as an internal state of an individual’s central nervous system, elicited by instrumental reinforcers, that gives rise to physiological, behavioural, and cognitive responses. In these regards, instrumental reinforcers are elements an animal will either work for (positive) or avoid (negative) [7]. Indeed, emotions set a mood that regulates the perception of incoming stimuli [8]. As a consequence, emotions are in control of what the individual regains in its memory, shaping the processing of information and the accuracy of conclusions [9,10,11]. Contextually, mood has been defined as a relatively enduring affective state that arises when an experience in one context modifies the individual’s reaction to future events [12].

It is in the framework of the emotional background of individuals, in which affect occurs [7]. Affect refers to an experienced emotion, which can be defined as a stimulus-directed affective state, and consists of behavioural, physiological, and cognitive components. Furthermore, affect may occur outside of awareness [4]. Studies on a variety of non-human species indicate that, as in humans, pessimism may be related to negative affect or mood [8], even if the conscious experience of such a state cannot be known for sure. While individuals reporting negative affective states, for instance, depression or anxiety, tend to make negative judgements about the future, and are commonly referred to as “pessimistic” [9], “optimistic” individuals may rather report positive states linked to higher extraversion (agreeableness, conscientiousness, and openness) and lower neuroticism [10].

Early studies on cognitive bias, in non-human animals, focused on the comparison of the outcomes of individual coping in different environments and the animal’s response to ambiguous stimuli [7]. Harding, et al. [11] concluded that individuals living in unpredictable environments tended to display rather pessimistic statuses, through behavioural patterns which are indicative of negative bias, such as fewer and slower responses to ambiguous stimuli [7]. Likewise, Burman, et al. [7] found that individuals held in blank environments responded with less excitement to an ambiguous stimulus, when their responses were compared to those of subjects being held with enriched surroundings.

Analogous studies have associated negative affective states with negative cognitive bias in non-human animals. However, in these cases, behaviours have been deemed harder to interpret as there is a lack of positive affective state measures and many of the criterion lack a priori hypotheses for how these may change depending on an individual’s emotional state [5]. Nonetheless, cognitive bias tests have been set as a very reliable and non-evasive indicator of animal welfare for several mammals [12], among which equine congeners can be found [13], as well as avian [14] and invertebrate species [15].

The application of judgement bias tests have been used in emerging research aiming to deepen the understanding between the potential connection of the function of cognitive bias with personality in human and non-human animals.. Réale, et al. [16] defined animal personality as the differences between individuals’ average level of behaviour that are repeatable across time and contexts.

Such time- and context-consistent individual personality differences may be a result of the highly complex dimensionality of personality itself. Contextually, the diverse nature of personality may possibly be linked to the fact that the expression of certain features of personality may be more prevalent over others across individuals and thus, such unequal expression has the potential to bias individuals’ judgements. For instance, studies have suggested more social, extraverted, proactive, and less exploratory individuals may potentially judge ambiguous situations in rather positive manners [17].

Finding evidence of a personality component to cognitive bias may shed some light on the level at which individuals may be bound to be optimistic or pessimistic, independent of their previous lifelong circumstances.

Personality features may not only constitute as a source of judgment bias variability. In this context, despite the existence of sex-mediated differences being often addressed, the causality for such differences has been largely overlooked. In these regards, Barker et al. [18] provided certain evidence that differences between sexes may actually derive from the existence of a greater intra-individual variability in females, as supported by their study in rats. These authors addressed the cyclic release of hormones as part of the oestrus cycle to be the most likely cause for this broad variability, which indeed has been supported by psychiatric literature. For instance, some infrequent disorders may parallelly appear during the conditioned course of ovarian cycle phases, beginning just after puberty and subsiding right after menopause [19]. Still, the aetiology of judgment bias may involve the interaction between environmental factors more than the limited influence of factors separately.

Contextually, the effects of sex may be better understood in the framework of other factors such as age. Age-dependent variability in judgment bias may be ascribed to the considerable plasticity which characterizes brain development along the life of individuals. Furthermore, the activation or maturation of certain regions of the brain, such as the cortico-limbic system, which have been proven to be responsible for the expression of emotions and hormonal responses to stress, may be enhanced or inhibited by situations that the individual experiences as distressing [20].

Although cognitive bias has attracted researchers’ interest for decades, related questions have only been recently approached in equids [21], and very briefly in donkeys [13]. In these regards, research has concluded that horses show variable judgement bias when housed in either restricted or naturalistic situations [22], and when housed either in stalls or herds [23].

In the case of donkeys, discrimination learning abilities have been suggested [24] but the only reported results for judgement bias tasks in this species are those by McGuire et al. [13], who comparatively evaluated the performance of horses and donkeys, to conclude rescued individuals were more optimistic than non-rescued individuals [13].

The wide variability and particularities of donkey breeds [25] has promoted the adaptation of cognitive tests used in Miniature donkeys [26]. However, no evidence for interindividual and interbreed variability components have been reported, even if there is a strong possibility that selective breeding has led to temperamental and cognitive differences [27,28].

Consequently, the aim of this study was to determine evidence of interindividual variability in Miniature donkeys in their judgement bias upon an ambiguous stimulus, whether biases may ascribe to personality factors or whether other factors such as sex, age or weather elements may be involved. The present research may not only improve the understanding of the levels at which individuals living under self-desirable conditions might experience negative mental states and low expectations towards new stimuli, but also, the ability to predict that an individual might experience negative emotions more often than others based on aspects of personality. This knowledge may offer a new insight into animal interactions with the environment, which may provide relevant information for caretakers who deal with animals on a daily basis and may be crucial for animal welfare studies aiming to disentangle the complexities of animal–human interaction, as it has been reported for decades in other species in which greater optimism has been reported under conditions which had indirectly been deemed to be indicative of better welfare [14,29].

## 2. Materials and Methods

### 2.1. Study Subjects

Twelve (*n* = 8 females, 3 gelded males and 1 whole male) rescued donkeys ranging in age from one to twelve years (5.75 ± 3.89 years, mean ± SD) were tested. Equids were housed at the facilities in T.S. Glide Ranch in Davis, CA, USA (Google plus code: G5R7+9P Davis, CA, USA/coordinates: 38.5409804, −121.8378263). Housing enclosure comprised a paddock (12 m^2^) shared with other donkeys, goats and cows. All donkeys were housed in the same facilities to ensure husbandry routines and environmental conditions were as similar as possible. Miniature donkeys were fed a diet of grain and hay once per day (between 7 h and 10 h in the morning) by staff at the facilities tailored depending on the individuals’ needs. Feeding was not adjusted before, during, or after the study. Handling practices did not involve human interaction apart from routine feeding and cleaning, occasional visits by the veterinarians and monthly blood collection for research purposes.

### 2.2. Subjects Historical Record

The Miniature donkeys used as subjects in this study were rescued in 2016 from two different private owners, and brought to T.S. Glide Ranch where the study was developed. Rescue ages varied with individuals variably being rescued as adults and foals. Two donkeys were born at the ranch in 2018. The donkeys had not undergone any prior systematic training previous to this study. Study subjects did not undergo any previous training in line with the execution of the test developed in the present research as certified in veterinary examinations. Complete clinical evaluation was performed prior to the testing experiences, and equids were classified based on their body score upon arrival. Tests did not start until all the animals involved had a healthy body score of 5, as it has been suggested for donkeys being transferred between housing facilities [30].

### 2.3. General Details

Maximum total length of the experiment per animal was 55 min. A total of 6 trials were performed per donkey and day.

### 2.4. Stage 1: Pre-Test Habituation

During the pre-test habituation phase, donkeys were brought into the testing area for 30 min a day. This stage was aimed at habituating the donkeys to the judgment bias set-up. Donkeys were given 20 min for the pre-test habituation phase and if they had not located the new objects, the test was terminated. Habituation sessions involved two operators (handler and operator) and additional donkeys to those being trained in the test arena, to promote social buffering [31] and social facilitation towards exploration of new objects [32,33]. Both operators had credited experience handling equids. The handler was blind to the assigned reward side for each equid and was responsible for handling the equids (transferring animals from housing facilities to experimental area and remaining motionless at the starting line within the area during the sessions). The second operator was responsible for the supervision of the sessions, filling the buckets, filming the trials, and recording the data (in case of video failure). Their roles and tasks were maintained across the three stages that the study comprised (Habituation/Training/Testing). Donkeys participated in a minimum of 3 habituation sessions. During this period, each individual was brought to the test arena for half an hour per day, led by the handler at a walk with a halter and lead line, and then released for the remaining time of each habituation session by unclipping a lead rope from the donkeys’ halter. Animals were released to allow them interact with, investigate, and become familiar with the test apparatus. After release, each individual had 20 min before the session was terminated. Once released, the handler stood motionless in the same position. The operator supervised the training sessions and annotated results or potential incidences from outside the experimental area.

### 2.5. Stage 2: Pre-Test Training (Detour Phase)

The same two operators (handler and operator) involved in habituation sessions were involved in the detour phase of the training sessions, but only one donkey at a time accomplished each training stage. Pre-test training took place in the same paddock as habituation.

During the detour phase, the animals learned to distinguish between a positive and a negative stimulus, in this case a black bucket full of food and a white empty one. The detour phase consisted of five positive and five negative trials and was repeated until the minimum number of repetitions for each individual to discriminate between stimuli was enough. Trials were presented in a pseudorandomized order with no more than 2 consecutive trials of the same type. The first trial of the first session for each donkey was loaded to encourage the donkeys to become interested in investigating the buckets. The side that was designated as the bucket end was counterbalanced across donkeys, with half of the donkeys learning the location on the right as the bucket end and the other half learning the location to the left as the bucket end as suggested in McGuire et al. [13]. Operators used the same steps when setting up a loaded and unloaded bucket. The only difference was that no food was left accessible after setting up the unloaded trials. Positive trials were characterized by the animals being presented a fully loaded bucket place at the opposite end of the area to which the individual was placed. By contrast, during negative trials the individuals were presented an empty bucket on the opposite end (A+ or A−, Figure 1). Only a bucket was placed within the test arena at a time. Once buckets had been placed on location + or −, the individual being tested was brought to point A (Figure 1). Afterward, the individual was released and allowed to explore and, in the case of positive trials, to feed.

Latency was measured and registered as the time spent between the donkey’s release at A and the moment when its throat latch was less than one metre away from the bucket. The aforementioned latency was chosen, as it was reported to be the most informative in similar studies in equids [13]. When after 3 min, the donkey had not reached the bucket, the handler guided it to the goal and revealed the bucket’s content. Trials were considered to be finished after 3 min or when the individual’s throat latch was less than one metre away from the goal. After each trial, during the intertrial interval, while the handler brought the donkey back to the start (location A), another operator swapped the buckets so that the set up was correct for the following trial. All donkeys were led away from the buckets area and oriented in the opposite direction during the intertrial intervals while buckets were being swapped to prevent the animals from reliably using olfactory, auditory, or visual cues (watching the operator loading the bucket) to enable them to issue the correct judgment. Inter-trial intervals lasted approximately 2 min.

Each set was composed of three positive and three negative trials, randomly ordered in a way that no more than two of the same trials followed each other (i.e., + + − − + −, + + − + − −, − + + − − +, + − − + + −, − − + + − +, − + − − + +) and the animals rested for twelve hours between sets. Donkeys were considered fit to move on to the treatment phase when the latency to reach the positive goal was significantly shorter than the latency to reach the negative goal (Mann–Whitney test, *p* < 0.05).

Food buckets were located at the same place as they would be during the testing sessions to habituate the subjects to eating from the bucket. The food bucket did not have a lid during the whole experience, as this has been reported to increase fear without decreasing latency when developing the task to seek for the rewards (food), hence potentially influencing the individuals’ choices in turn [34]. The animals were fed prior to the tests, which they were accustomed prior to the study. A food mix of apples, carrots, dry treats, and grain was used to appeal to and cover individual preferences. Food was used as a motivator to encourage donkeys to complete the tasks in all experiments [35]. Afterward, the donkeys went through three weeks of training, with 1 training session per day, to increase cooperation with the operators and reduce fear levels associated with the test arena and material following the premises described in Christensen et al. [34]. Although reminders of habituation have been performed in other equine species such as horses [13], habituation was not performed after training in the present study as animals had not lost prolonged contact or exposure to the instruments used for the study until it was accomplished.

### 2.6. Stage 3: Judgement Bias Testing (Treatment Phase)

Once the habituation and pre-test training (detour phase) had been fulfilled, the donkeys were tested for judgement bias, adapting the protocol by Freymond et al. [36]. All individuals were tested for judgement bias at the same time. Food rewards were deposited in the bottom of the corresponding bucket 1 min before the start of each session so that scent cues were controlled to make sure subjects used only location cues to issue judgements. The same two operators of the previous stages participated and maintained their tasks during judgement bias testing (treatment phase). The order in which the animals performed the treatment phase was always randomized.

The treatment phase exposed the animals to three new buckets of different colours: dark grey, aligned to the right of the black bucket’s location (+A); medium grey, aligned to an intermediate position between the black and the white bucket’s location (An); light grey, aligned to the left of the white bucket’s position (−A) (Figure 1). These were presented to the test subjects individually, always empty and in random order, across two sets of seven trials (+ − A+ + An – A−; − + A− − An + A+). Individual exposure to this phase was limited to two sets, in order to avoid learning [36,37]. The remaining steps of the experimental procedure were the same as those described for the detour phase. All individuals completed judgement bias testing.

### 2.7. Experimental Area

Research took place in an outdoor experimental arena which was 20 m away from the animals’ separate housing enclosure. A schematic depiction of experimental area is shown in Figure 1.

The experimental area was 12.8 m × 16.8 m, thus 215.04 m^2^. The starting line for the equids was 12 m away from the line where buckets were located. Bucket line was located 3 m away from the bottom of the experimental area and consisted of 5 locations at which one bucket was placed. Each bucket/location was 1 m apart. Buckets were not stacked together.

### 2.8. Personality Evaluation

Individual personality was scored using an adaptation of the questionnaire proposed by Navas et al. [38], and French [39]. The aforementioned methodology has been previously used to evaluate behavioural patterns in problem-focused coping styles, the genetic background behind them and the conditioning factors involved in donkeys [24,27,28,35,40]. A questionnaire was completed by ten observers. Observer sample comprised people whose frequency of contact with donkeys varied from observers in contact with the donkeys in the study on a daily basis (3 caretakers), observers who had worked with the animals for the period of five weeks comprising habituation and training (five operators) and observers in occasional contact of at least two times a week with the donkeys (2 care assistants). The expression degree of a total of eighteen behavioural traits was ordinally rated from 1 to 10 with these values representing the minimum and maximum extremes in the scale, respectively. The description of each of the levels comprising each of the traits in the scale was accessed from Navas et al. [35] and Navas González, et al. [40], and a summary of the descriptors, scales, and extreme adjectives used is shown in Table 1.

### 2.9. Data Analysis

#### 2.9.1. Previous Assumption Testing

Since sample size was a limitation in this study, parametric assumptions were tested to decide on the most appropriate statistical approach to follow. The Shapiro–Francia W’ test (for 50 < *n* < 2500 samples), Shapiro–Wilk test (for *n* < 50 samples), and Levene’s test were used to discard gross violations of parametric assumptions (normality and homoscedasticity). The Shapiro–Francia W’ test was performed using the Shapiro–Francia normality routine of the test and distribution graphics package of the Stata Version 15.0 software. A gross violation of normality assumption occurred in all variables (*p* < 0.05). Homoscedasticity was violated as well (*p* < 0.01); hence, a nonparametric approach was suggested. Homoscedasticity was tested using the explore procedure of the descriptive statistics package in SPSS Statistics (Version 25.0, IBM Corp., Armonk, NY, USA).

#### 2.9.2. Inter-Rater and Intra-Rater Reliability

Intra-class correlation coefficient (ICC) analysis was performed to confirm intra-observer reliability, among those who answered the personality questionnaire. For each individual, an average was calculated of the scores given by all the observers in each question. Intra-class correlation coefficient analysis was performed using the Scale routine of the Reliability analysis procedure of SPSS Statistics for Windows, version 25.0, IBM Corp. (Armonk, NY, USA) [41]. As suggested in Koo and Li [42], and considering the present study conditions, ICC values less than 0.5 are indicative of poor reliability, between 0.5 and 0.75 indicate moderate reliability, between 0.75 and 0.9 indicate good reliability, and greater than 0.90 indicate excellent reliability [43].

#### 2.9.3. Adjusted Latency Calculation

Data from the judgement bias tests was averaged to an adjusted latency (Adj.L) to approach the ambiguous stimulus (An), for each donkey [8]. The latter was calculated considering the individual’s latencies to approach the positive and negative locations, as follows:(1)$$\mathrm{Adj}.L=\frac{\bar{X}\;\mathrm{latency}\;\mathrm{to}\;\mathrm{ambiguous}\;\mathrm{stimulus}\;(\mathrm{An})-\bar{X}\;\mathrm{latency}\;\mathrm{to}\;\mathrm{rewarded}\;\mathrm{stimulus}\;(A+)}{\bar{X}\;\mathrm{latency}\;\mathrm{to}\;\mathrm{unrewarded}\;\mathrm{stimulus}\;(A-)-\bar{X}\;\mathrm{latency}\;\mathrm{to}\;\mathrm{rewarded}\;\mathrm{stimulus}\;(A+)}$$
With $\bar{X}$ being the mean.

Donkeys were considered fit to move on to the treatment phase when the latency to reach the positive goal was significantly shorter than the latency to reach the negative goal (Mann–Whitney test, *p* < 0.05).

Adj.L was considered the continuous dependent variable on which to perform the Mann–Whitney’s U test and Spearman’s rank correlation analysis to test for the effects of the independent variables of personality features, weather and daytime.

#### 2.9.4. Personality Questionnaire Analysis: Judgement Bias Links

Spearman’s rank correlation coefficient was tested for two different aims. First, to detect the relationships between personality questionnaire questions to ascribe the questions to broader personality features.

Second, once significant correlation between question pairs had been identified, the scores of the correlated questions were averaged to calculate a score for each personality feature to determine the links between judgement bias and these personality features. Spearman’s rank correlation coefficients were calculated across question pairs and between adj.L and personality feature scores using the *Bivariate* routine of the Correlate procedure of the Analyze set in SPSS Statistics for Windows, version 25.0, IBM Corp. (Armonk, NY, USA) [35].

#### 2.9.5. Effects of Weather and Daytime

To assess the effect of weather on judgement-bias-related latency, the variables of relative humidity (%), wind (m/s), and temperature (Celsius degrees) were considered. Additionally, the presence of differences in the median latency as a potential effect of daytime (morning or afternoon) was also evaluated. To this aim, the relationship between adjusted latency and weather variables was explored calculating the Spearman’s rank correlation coefficient of the *Bivariate* routine of the *Correlate* procedure of the *Analyze* set in SPSS Statistics for Windows, version 25.0, IBM Corp. (Armonk, NY, USA) [35]. Afterwards, Mann–Whitney’s U test was performed to detect differences in the mean latency between morning and afternoon testing moments (Daytime). Mann–Whitney’s U test was performed using the 2 Independent Samples approach of the *Legacy Dialogs* pack of the Nonparametric Tests routine of the Analyze procedure in SPSS Statistics for Windows, version 25.0, IBM Corp. (Armonk, NY, USA) [35].

## 3. Results

### 3.1. Inter-Rater and Intra-Rater Reliability Analysis

An excellent (ICC > 0.9) degree of inter- and intra-observer reliability was confirmed for the scores from personality questionnaires within and across observers.

### 3.2. Judgement Bias

During detour phase, the group average latency to approach the positive stimulus (A+) was 30.3 ± 34.4 s, and 77.2 ± 65 s to approach the negative stimulus (A−). During treatment phase, the mean latencies to approach the stimuli were 33.7 ± 43.1 s (positive), 37.9 ± 50 s (A+), 85 ± 70.3 s (An), 91.5 ± 71.4 s (A−) and 145.5 ± 53.1 s (negative) (*n* = 12). Latencies to approach the ambiguous stimulus (An) varied between individuals and were plotted in Figure 2.

### 3.3. Personality Questionnaire Analysis

#### 3.3.1. Correlations across Personality Features

Table 2 reports the relationships that exist across the personality features evaluated through the personality evaluation questionnaire.

#### 3.3.2. Links of Judgement Bias with Personality Features

The presence of significant Spearman’s correlation between adj.L and some personality features, hence, differences in approach latency depend on the degree of expression of certain personality features. Contextually, a significant positive Spearman’s rank correlation was found between patience and negative judgement bias (rs = 0.714, *p* < 0.01). No significant correlations were found between negative judgment bias and the remaining personality factors (Spearman’s Rank correlation, *p* > 0.05).

### 3.4. Sex, Age, Historical Record, and Weather Effects

The environmental factors registered during the tests—temperature, wind, and humidity—did not have a significant effect on the tests’ results (Mann–Whitney test, *p* > 0.05) (Table 3). The individuals’ life history (whether they were rescued as adults, rescued as foals or born at the ranch) did not significantly influence the variables under study (Kruskal–Wallis test, *p* > 0.05). Sex effects were not significant (Mann–Whitney test, *p* > 0.05).

## 4. Discussion

Sample size limitations compel the nature of the present study to be a preliminary explorative/descriptive approach to the understanding of judgment bias in Miniature donkeys. In this context, considering more animals may have been the preferable option to increase statistical power, this is not always feasible in animal studies due to availability, endangerment status of the animals, or ethical reasons [44]. This may in turn compromise the replicability thus, the validity of studies. However, certain alternatives can be used to improve the detection of true significant effects and correct estimation of the size of these effects in reduced sample size contexts when the possibility to increase the number of participants is not feasible in a study.

In these regards, Meyvis and Van Osselaer [45] suggested that by taking the appropriate measures prior to running an experiment, and taking the pertinent post experiment statistical interpretation cautions, researchers may somehow counteract the detrimental effects of small sample size on observed effect sizes increasing the ability of tests to detect significant factor effects.

Contextually, the same authors suggested that when individualizing variables (those linked to capture individual variability) as covariates, the sample size required to detect the effect with adequate power is halved. This hypothesis has been supported in behavioural studies in donkeys [35], which reported reliance on several factors individually may help quantify the factors or effects involved more accurately than their conjoint effects (interactions), mostly based upon the fact that considering interactions in pairwise comparisons so numerous that one sample may not be able to hold for them.

Additionally, the better performance obtained after the separate evaluation of effects has been ascribed to the fact that conjoined evaluation of effects may lead to confounding interactions between variables [46,47,48]. These confounding variables play their part in effect overinflation, which otherwise is buffered when potentially confounding variables are tested separately. Further, this may prevent the fact that collateral relations between personality characteristics and age (potentially also sex) may have been evaluated even subsequently, and given this may lead to an enormous number of tests, with consequent inflation of the Type I error.

However, in small sample size contexts, if a sample is small, the finding of no effect of a factor on a variable when indeed there is one (Type II error, false negative), may be more frequent than finding an effect when there is not (Type I error, false positive). The benefit from this is, on the other hand, that it may generally lead to less serious consequences than those derived from a Type I error (false positive). This means that when an effect is detected, there is an effect, although it is true that if sample was larger, additional significant effects (small and medium effects to be detected) or even a different magnitude (small effects to be medium or large in reality) for the same effects could be found.

The emerging study of intrinsic individual differences in judgement bias—possibly linked to well-known personality traits—is of great importance to understand animal welfare, and to further develop animal behaviour studies. Emotions have been suggested to have developed to meet different environmental demands, leading to links with different cognitive methods [49]. Indeed, it is the manner in which a certain individual faces or confronts the challenges meant by environmental demands (optimism vs. pessimism) which may condition the animal’s emotional status after an expected outcome has or has not been achieved [50]. For instance, an animal may consider itself to be more capable of performing a certain task or to achieve a reward than it may physically be prepared for, which can often translate into them being frustrated [39] or making poor choices that lead to physical harm [51].

In turn, it is the balance between what the individual expects and the chances of these expectations to be fulfilled which may determine the long-lasting effects or the discard of the procedures implemented to respond to the aforementioned environmental demands. In this context, the term cognitive bias has been suggested to be linked to an irrational decision-making process, often associated with detrimental emotional states. As a result, a biased decision may prove itself adaptive if taken into consideration with the information on which the decision is based [5].

As previously reported by McGuire et al. [13], the donkeys in the present study learned to discriminate between a positive and a negative stimulus. In this regard, as the only motivating element presented was food, the removal of food in the bucket was proven to be enough to promote discrimination learning, deeming it unnecessary to use negative reinforcement for this kind of test. The use of negative reinforcement has been reported to be less effective than positive or even neutral reinforcement when promoting learning abilities in donkeys, which may support our findings [24].

The time spent at the positive location during negative and ambiguous trials suggested that spatial location cues may play a more important role than colour discrimination. Though horses have dichromatic vision and are able to distinguish brightness levels [51], additional studies would be necessary to confirm that colour discrimination was even incorporated in the study of donkeys’ cognitive processes to solve this task. Individuals differed from each other in their judgement bias, as exhibited in their different latencies to approach an ambiguous stimulus. Since there were no treatment differences, the variation found between individuals must be attributed to either life history, current housing situation, personality, or a combination of these, as reported by other authors [52].

The positive correlation found between patience and the judgement bias score is indicative of the fact that the individuals classified as more patient were also more pessimistic. This finding contrasts with information reported by authors such as Arnout and Bedair [53], who detected a highly significant correlation between patience and pessimism in humans. This disagreement may rely on the fact that patience, however, as interpreted by the observers, may instead reflect reduced nervousness or low exploration rates.

Additionally, this personality factor may reflect that individuals are generally slow when solving new challenges, as observed during the detour task. This could be supported by the research developed by several authors [21,54], who suggested the fact that although donkeys reported the highest rates of success at cognitive tests, the latencies needed to solve the problems presented was longer than that reported by mules. From a different perspective, expectations towards some stimuli may perhaps reduce the individual’s motivation to interact with them, presenting as passive behaviour we understand as patience.

The latter has been studied as an adaptive choice of renouncing immediate benefits to acquire more valuable future rewards [55]. This behaviour may not only be related to the expectation of a better future reward but also, in some cases, with the expectation of no reward at all. Moreover, if we assume that individuals with generally lower activity levels may have been classified by the observers as more patient, this could partially explain why they react less promptly to an unfamiliar stimulus.

An underlying link between personality traits related to interactions with unfamiliar environments, and the individuals’ expectations towards unknown situations might be present, as observed in pigs during a social isolation and a novel object task [56]. The fact that donkeys show different judgement bias elements, which are not correlated with their life history or living conditions, indicates that optimism/pessimism might not always be a consequence of acute stress or inappropriate welfare. Instead, it suggests that a trait-like feature might be conditioning these behaviour patterns. It is important to define how these biases affect animals’ behaviour and cognition, in order to improve our interaction with them and to broaden our notion of animal wellbeing.

## 5. Conclusions

Spatial location cues may play a more important role than colour discrimination. The variation found between individuals must be attributed to either life history, current housing situation, personality, or a combination of these. A patent connection between patience and pessimism may occur in donkeys, which contrasts the findings in other species. Differences may derive from the fact that patience, however, as interpreted by the observers, may reflect reduced nervousness or low exploration rates. Individuals are generally slow when solving new challenges. From a different perspective, expectations towards some stimuli reduce the individual’s motivation to interact with them, generating the passive behaviour we understand as patience. The fact that donkeys show different judgement bias elements which are not correlated with their life history or living conditions, indicates that optimism/pessimism is not always a consequence of acute stress or inappropriate welfare. Instead, it suggests that a trait-like feature might be conditioning these behaviours.

## Figures and Tables

**Figure 1 animals-11-02737-f001:**
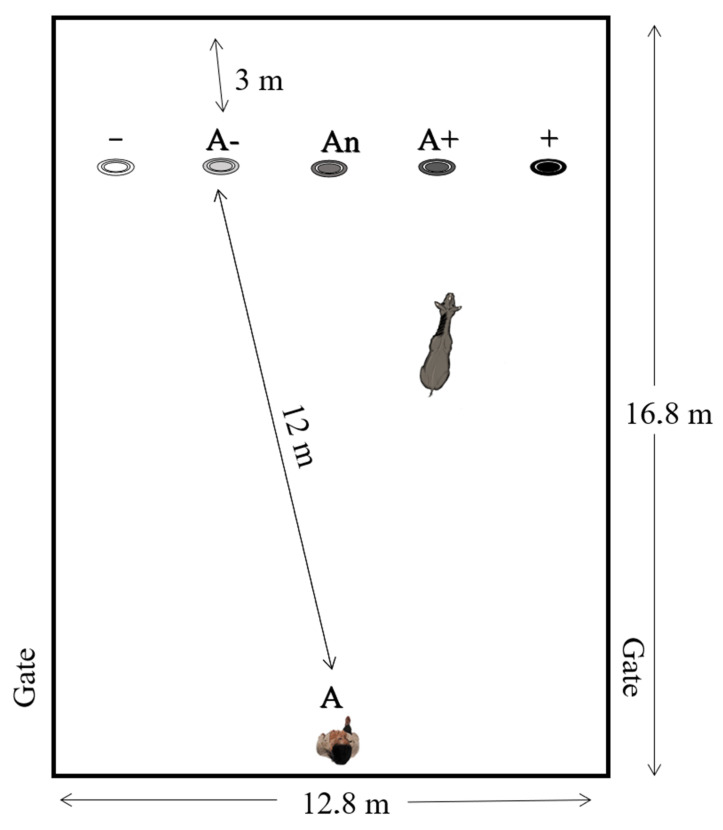
Experimental area. Experimental set-up during judgement bias trials. Position of the positive goal (+; right or left depending on the donkey, rewarded), the negative goal (−; opposite side as the rewarded one, unrewarded), the ambiguous goals (A−, An, A+; equidistant angles between the positive and negative buckets), the start (A) and entrance gates. The latency to reach the goal was measured as the time to go from A to one metre away from the tested bucket.

**Figure 2 animals-11-02737-f002:**
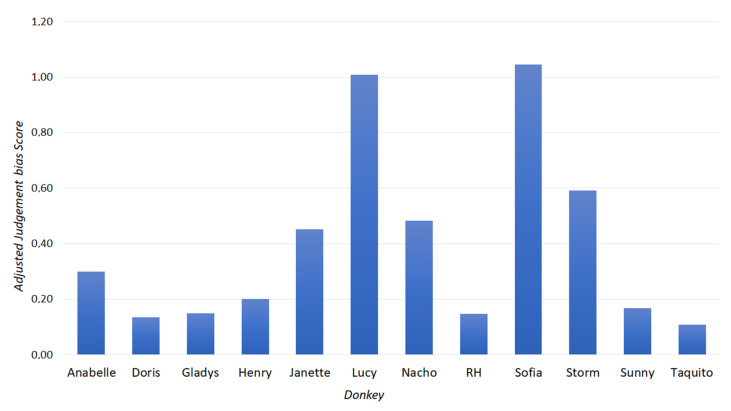
Adjusted judgement bias score (calculated mean latencies to approach the ambiguous stimulus—An, during treatment phase of the judgement bias test). Individuals that approach the stimulus more promptly have lower scores.

**Table 1 animals-11-02737-t001:** Summary of personality trait descriptors used in the testing stage of the judgement bias test, score ranges/scales, trait extremes and the references from which these were accessed.

Trait/Extreme Pair	Trait Extremes	Description	Extreme Score	Score Range	Reference
Calmness/Nervousness (Agitation)	Calm	Does not get startled. Pays attention to other stimuli around at the same time that it pays attention to problem stimulus. Does not show interest in moving toward the problem stimulus, but does not avoid it.	1	1–10	[28,39]
	Nervous	Gets startled. Rejects moving towards the problem stimulus. Rejects moving towards the problem stimulus.	10		
Focused/Unfocussed (Concentration)	Focused	Does not get startled. Only focuses on the problem stimulus. Does not get distracted with the environment.	1	1–10	[27]
	Unfocussed	Gets startled. Gets distracted with the environment.	10		
Dependent/Independent (Group Dependency)	Dependent	Comfortable and displays a quiet mood when separated from the herd or when in an unfamiliar environment.	1	1–10	[27,28]
	Independent	Is not comfortable. Restless when separated from the herd and/or when in an unfamiliar environment.	10		
Untrainable/Trainable (Trainability)	Untrainable	Incapable of learning to fulfil the test and/or does not respond promptly.	1	1–10	[27]
	Trainable	Very easy to teach to fulfil the test, responds promptly.	10		
Easily Excitable/Unexcitable (Excitement)	Easily Excitable	Gets startled and moves towards the problem stimulus. May get distracted with the environment.	1	1–10	[27,28]
	Unexcitable	Does not get startled but only focuses on the problem stimulus. Does not get distracted with the environment.	10		
Unfriendly/Very Friendly (People Friendliness)	Unfriendly	Unwilling or reluctant to human contact or when operators approach.	1	1–10	[27,28]
	Very Friendly	Very willing and proactively seeking human contact or approaching operators.	10		
Unconcerned/Curious (Curiosity)	Unconcerned	Does not get startled. Stands still or rarely approaches the problem stimulus. Gets distracted with the environment.	1	1–10	[27,28]
	Curious	Does not get startled. Only focuses and completely and promptly approaches problem stimulus.	10		
Short Memory/Exceptional Memory (Memory)	Short Memory	Does not remember stimulus or previous learning.	1	1–10	[27,28]
	Exceptional Memory	Perfectly remembers stimulus or previous learning.	10		
Stoic/Panicking (Fearfulness)	Stoic	Does not get startled. Shows an indifferent attitude towards the problem stimulus. It may approach it, but does not hold interest.	1	1–10	[27,28]
	Panicking	Gets startled. Only focused on the stimulus being presented. Tries to move away from the stimulus presented, but if it does not succeed, it stands still. Does not move towards the problem stimulus if led by the operator.	10		
Uncooperative/Cooperative (Cooperation)	Uncooperative	Does not cooperate with the operator during their interaction.	1	1–10	[27,28]
	Cooperative	Easily cooperates with operator during their interaction.	10		
Unpredictable/Predictable (Emotional stability)	Unpredictable	Its reactions are not predictable.	1	1–10	[27,28]
	Predictable	Its reactions are predictable.	10		
Stubborn/Compliant (Obstinacy)	Stubborn	Systematically rejects or react oppositely as it should.	1	1–10	[27,28]
	Compliant	Systematically does what it is intended to do without rejection.	10		
Alert/Dull (Vigilance)	Alert	Shows an alert status and focuses on the problem stimulus.	1	1–10	[27,28]
	Dull	Shows an apathic status and does not focus on the problem stimulus.	10		
Impatient/Patient (Perseverance)	Impatient	Impatient in the course of the test.	1	1–10	[27,28]
	Patient	Patient in the course of the test.	10		
Submissive/Dominant (Congeners Competitiveness)	Submissive	Shows a submissive attitude in the presence of congeners.	1	1–10	[27,28]
	Dominant	Shows a dominant attitude in the presence of congeners.	10		
Impassive/Surprised (Surprisability)	Impassive	Does not get startled. Shows an indifferent attitude towards the problem stimulus. It may approach it, but does not show interest.	1	1–10	[27,28]
	Surprised	Gets startled, but progressively relaxes. Only focused on the stimulus being presented.	10		
Shy/Outgoing (Shyness)	Shy	Does not enter the testing area confidently. Approaches problem stimulus but presents a shy attitude, although interest is present.	1	1–10	[27,28]
	Outgoing	Confidently enters testing area. Approaches problem stimulus steadily and straightforward.	10		
Easy/Difficult to handle (Housing Entering/Leaving)	Easy to handle	Easily enters/leaves housing facilities.	1	1–10	[27,40]
	Difficult to handle	Reluctant to enter/leave housing facilities.	10		

**Table 2 animals-11-02737-t002:** Spearman’s correlation coefficients (rs) between questions of the personality questionnaire (Q1–18). Scores for each question were calculated as the average between all individual scores (mean score of the ten observers), *n* = 121 observations.

Questions																			
**Q1**	**Q2**	**Q3**	**Q4**	**Q5**	**Q6**	**Q7**	**Q8**	**Q9**	**Q10**	**Q11**	**Q12**	**Q13**	**Q14**	**Q15**	**Q16**	**Q17**	**Q18**		Questions
Agitation	Concentration	Group Dependency	Trainability	Excitement	People Friendliness	Curiosity	Memory	Fearfulness	Cooperation	Emotional stability	Obstinacy	Vigilance	Perseverance/Patience	Congeners Competitiveness	Surprisability	Shyness	Housing Entering/Leaving		
	−0.490	−0.242	0.025	**0.768 ****	−0.576	0.023	−0.576	**0.622 ***	−0.229	0.236	0.109	0.394	−0.468	**−0.607 ***	**−0.595 ***	**−0.756 ****	**0.852 ****	Agitation	Q1
		0.298	0.183	**−0.579 ***	**0.676 ***	**0.620 ***	0.366	**−0.640 ***	0.513	−0.056	−0.227	−0.402	0.224	**0.688 ***	0.402	**0.704 ***	**−0.620 ***	Concentration	Q2
			0.502	−0.455	0.239	0.235	0.022	−0.511	0.412	−0.325	−0.204	−0.396	0.244	0.024	0.164	0.492	−0.476	Group Dependency	Q3
				−0.192	0.206	0.411	0.206	−0.389	**0.676 ***	0.137	−0.174	0.083	0.141	0.101	0.013	0.232	−0.129	Trainability	Q4
					−0.556	−0.133	−0.534	**0.631 ***	−0.449	0.491	−0.199	0.484	−0.275	**−0.673 ***	−0.354	**−0.828 ****	**0.778 ****	Excitement	Q5
						0.619*	0.214	**−0.766 ****	0.530	−0.300	0.131	−0.025	0.379	0.395	**0.862 ****	**0.745 ****	**−0.619 ***	People Friendliness	Q6
							0.095	−0.406	0.481	0.238	−0.122	−0.076	0.320	0.163	0.352	0.364	−0.214	Curiosity	Q7
								−0.451	0.337	0.110	−0.383	−0.259	0.013	0.395	−0.004	0.300	−0.357	Memory	Q8
									**−0.779 ****	0.435	0.181	0.321	−0.179	−0.308	−0.580 *	**−0.751 ****	**0.721 ****	Fearfulness	Q9
										−0.184	−0.194	−0.112	0.192	0.329	0.180	0.434	−0.481	Cooperation	Q10
											−0.557	0.171	0.331	0.000	−0.280	−0.454	0.286	Emotional stability	Q11
												0.110	−0.177	0.000	0.190	0.175	0.122	Obstinacy	Q12
													−0.181	−0.228	−0.024	−0.501	0.497	Vigilance	Q13
														0.358	0.422	0.340	**−0.602 ***	Perseverance/Patience	Q14
															0.156	**0.609 ***	**−0.675 ***	Congeners Competitiveness	Q15
																**0.644 ***	−0.512	Surprisability	Q16
																	**−0.854 ****	Shyness	Q17
																		Housing entering/leaving	Q18

Significant correlations are marked in bold as follows; *: *p* < 0.05 and **: *p* < 0.01.

**Table 3 animals-11-02737-t003:** Standard means and deviations for weather variables during the experimental procedures.

	Temperature (°C)	Wind (m/s)	Humidity (%RH)
Judgement bias (detour phase)	19.00 ± 5.11	0.48 ± 0.78	49.02 ± 11.00
Judgement bias (treatment phase)	15.08 ± 4.80	0.64 ± 0.94	55.84 ± 30.86

## Data Availability

Data will be made available from corresponding authors upon reasonable request.

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
