# Peer review of "Judgement Bias in Miniature Donkeys: Conditioning Factors and Personality Links"

_animals, 2021, doi:10.3390/ani11092737_

Round 1

Reviewer 1 Report

Following the last revision by the authors, the paper has improved a lot. Now the description of the experiment and the presentation of the results are clearer and easier to follow.

I appreciate that the authors explicitly recognise the limits of the studies in the Discussion (Lines 434-470; about this I would suggest they keep the whole part in a separate subsection after the discussion of the actual results). In particular, because they underline the "explorative/descriptive" nature of the study and warn about the interpretation of the results.

I just want to collect some general thoughts about the statistical approach. 

Overall I agree with the decisions and arguments of the authors, they used nonparametric statistical methods and evaluated separately the relation between each predictor and the outcome due to sample size limitation. However, I find all the following discussions about testing factors individually, issues related to confounding factors, interaction etc. to be overwhelming. It seems like this is the best approach when I would describe it as "the least worst". The meaning is the same but it affects how people would interpret the results. I want to underline this is not a critique of the authors, they did the most reasonable thing that was possible to do and I agree with them.

In general, I would recommend testing all predictors in the same model to evaluate the unique contribution of each predictor, but given the sample size, this is not even possible (predictors are greater than observations). So the approach adopted by the authors is the only one possible and is reasonable.

The problem, however, is what kind of inference can you make from this data? To be honest, in this study there were only 12 observations and 23/24 predictors were evaluated (plus correlation between predictors - in the previous version the correlation between Age and Vigilance was reported but now is no longer reported). Basically, everything was tested and predictors were selected according to statistical significance. Applying statistical significance filter in this scenario (extremely low power) leads to unreliable results both considering variable selection ad effect size estimation. As pointed out by the authors, false negatives are very likely as well as false positives (considering the number of tests 23/24 and no control of alpha-value it should be around 70%). Moreover, do you think a correlation of r = .714 to be a reasonable value? I don't know, I am not an expert in animal behaviour but it seems an extremely high value.

I mean there is a high risk that results are not reliable. The resulting association between patience and the outcome is almost surely an overestimation of the actual effect and it is likely that also other variables are associated with the outcome. From an extreme perspective, I would say that descriptive statistics are more informative and inferential statistics should be avoided (of course I know this is not practical for publication).

So finally we could question: is it worth it? Yes of course it is. Collecting data (especially with animals) takes a lot of effort, the important thing is just to understand what kind of claims data can support. Thus, it is fundamental to recognise the limits and be cautious in the interpretation of the results (as the authors did). 

Few other answers about the authors' comments in the rebuttal letter:
1. "I would ask the reviewer, do we believe everything that data confess? [...]" - I would say that it is important not only to choose the most appropriate statistical methods but also to be reasonable in the interpretation of the results. Considering the present results, given the sample size and the "test all approach" I would hardly believe something. These are explorative/descriptive results (as the authors underlined). 

2. "We understand what the reviewer stated. However, we have some reservation to apply cross validation techniques in the context of bivariate correlations. [...]" - My fault, I wasn't clear enough. When I suggested a Leave One Out sensitivity analysis, I was not referring to cross-validation (I mean there are no models to compare in the ability to predict new data). Rather, I was suggesting a kind of sensitivity analysis to evaluate possible influential cases and estimates variability. If removing one observation at a time leads to similar results, this would strengthen the results. But probably it is not something popular (I have seen it mainly in meta-analyses context).

3. "Actually, there was a misunderstanding. Although we did use the word cluster in this study what we did was to depict correlations using a dendrogram constructed using the construct unweighted pair-group method using arithmetic averages (UPGMA) Tree task from the Phylogeny procedure of MEGA X 10.0.5 to 364 display the correlations formed among personality questions." - If I am not wrong any tree or dendrogram used to classify/group something is a cluster. On Wikipedia they define UPGMA as "a simple agglomerative (bottom-up) hierarchical clustering method".

Main issues:

1. There are some incongruences in the Abstract/Data Analysis section/Results section on which variables were tested:
    - lines 383-389: "Additionally, the presence [?of?] differences in the median latency as a potential effect of day-time (morning or afternoon) was also evaluated." However, I could not find any mention in the result section (lines 425-430).
    - lines 425-430: results considering individuals’ life history and sex differences are reported but these tests are not mentioned in the Data Analysis section.
    - line 425: "Sex, age, historical record and weather effects". The results considering age is not reported in the Results section either mentioned in the Data Analysis section.

2. It is not clear to me if the 18 personality traits measured with the questionnaire or some kind of personality features/factors defined as a composition (average) of the personality traits were tested:
    - lines 370-375: "Spearman's rank correlation coefficient was tested for two different aims. First, to detect the relationships between personality questionnaire questions to ascribe the ques-tions to broader personality features. Second, once significant correlation between question pairs had been identified, the scores of the correlated questions were averaged to calculate a score for each personality feature to determine the links between judgement bias and these personality features."
    - Were composite scores created to summarise the 18 personality traits according to the statistically significant correlations reported in Table 2 and then tested? Given the previous discussion about sample size issues and reliability in the results, I find this approach to be not appropriate.

Minor issues:

1. lines 32-33: "A questionnaire evalu-ating eleven personality features [...]". Should be 18 personality features.

2. line 38: "(33.7±43.1 vs 145.5±53.1 seconds) (Mann-Whitney test, P<0.05Latencies si" I would put all in the same parenthesis and add the missing ")." To close the sentence

3. lines 83-84: "“pessimis-tic” [9], ‘optimistic’ " inconsistent use of " and '.

4. line 91: "stimuli [7]. Likewise, Burman, et al. [7]" I guess the first reference "[7]" should be removed as the previous sentence refers to [11]

5. lines 141-143: " McGuire, et al. [13], who comparatively evaluated the performance of horses and donkeys, to conclude rescued individuals were more optimistic than non-rescued individuals [13]." Remove final "[13]" as it is already cited at the beginning of the sentence.

6. line 327: caption of table 2 indicates "n = 121"  should be "n = 12"?

Reviewer 2 Report

I have no further comments and I accept the current form of publication. 

This manuscript is a resubmission of an earlier submission. The following is a list of the peer review reports and author responses from that submission.

Round 1

Reviewer 1 Report

The work is very interesting and brings new information for understanding not only the specifics of the species of donkeys, but can also contribute to a better understanding of the mechanisms of behavior in other species, including humans. The only technical disclaimer concerns the table (Spearman correlation) which should be better configured as it is not readable. The basic results that result from it should also be indicated.

Author Response

Reviewer 1

Comments and Suggestions for Authors

The work is very interesting and brings new information for understanding not only the specifics of the species of donkeys, but can also contribute to a better understanding of the mechanisms of behavior in other species, including humans. The only technical disclaimer concerns the table (Spearman correlation) which should be better configured as it is not readable. The basic results that result from it should also be indicated.

Response: We changed the layout of the table to include more information and make it more readable and understandable. Basic results were indicated from line 325 to 338.

Reviewer 2 Report

The topic of the study is interesting. The abstract is insufficient, it is not clear what the study is about and what the key findings are. Too many aspects are mentioned in insufficient detail.

The introduction needs a better link between the related topics investigated. The topic of judgement bias is introduced fairly well, personality research is briefly mentioned, but the link is not clear. Why should judgement bias be affected by personality? This needs to be made explicit based on theory. In the abstract, sex and age differences are mentioned. It needs to be explained how, based on theory, judgement bias is expected to differ depending on sex and age.

More details about the experimental procedure are necessary. How long have the experiments taken? In which order have the individuals been tested? How many tests per day were conducted? How much break between the tests? Please provide an overview over number of trials per individual. Have all individuals completed each stage of the experiment before moving on or have individuals been moved through experimental stages at different times? Have all individuals who started the experiment completed it? How long did it take each individual to complete each stage of the experiment?   

Details of analysis of judgement bias are missing. How was the data analysed?

The authors provide a very detailed analysis on personality clustering, effects of sex, age, weather. All these questions are not properly introduced, and the results section becomes quite confusing. Are these analyses relevant? What does this tell the reader? This is not a major issue and there are two potential fixes: (A) the manuscript could focus on the major research questions and all additional information and analysis can be moved into the supplementary materials; (B) all aspects of the analysis are properly introduced and explained to the reader throughout the manuscript.

Presently the presentation of the results can be improved. The authors need to go back to the main research question and provide 1-2 figures illustrating the results answering this research question. I have to admit, after reading the manuscript I am a bit confused what the actual research question is. Is it about judgement bias in donkey, about personality, about sex or age or weather effects? This needs to be made explicit. Not all factors need to be investigated but only the once for which clear predications can be derived from theory. There needs to be a clear quantitative analysis of personality and the personality aspect needs to be better linked to the judgement bias aspect. Are personality differences in judgement bias to be expected? Which differences, in which directions? Which personality aspects are expected to be related to an optimistic bias, which aspects are expected to be related to a pessimistic bias? Can this be confirmed by the present study? The results section is too descriptive and too little explorative. You need to tell the reader whether your results support your predictions or not, you cannot simply describe everything and leave it to the reader to figure it out.

Analysis: Could a multivariate approach be taken into consideration? As authors analyse factors like sex, age, weather, it seems appropriate to consider them in the statistical analysis. It seems inappropriate to analyse the effect of each of these factors independently and assessing the contribution to variance in the dataset of all the factors together seems appropriate.

Abstract:

Ln. 29: please mention who completed the personality questionnaire. Owners, caretakes, researchers? Please mention which experimental paradigm was used to test judgement bias

Ln. 33: ‘No significant differences were found between sexes (P>0.05).’: Without information on sample size and test used the p value is insufficient information. Either delete p value or provide additional information which would allow the reader to properly judge the information provided.

Ln. 33: ‘Vigilance behaviour significantly decreases as age progresses.’ Odd phrasing. I assume this is not a longitudinal study which followed individual donkeys across ages. Rephrase ‘vigilance behaviour significantly decreased with age’. Please put vigilance into context, why was the behaviour relevant, what do changes tell us?

Ln. 33-34: Which results provide this information? Please explain.

Ln. 35-36: How is patience related to this? Patience is a different cognitive concept compared to judgement bias. The authors do not explain which experiment they conducted and what they measured. More information needs to be provided in the abstract.

Ln. 38-41: Over-generalisation of results. Please provide a critical discussion on what the results of the experiments tell the reader (judgement bias, personality).

Laterality is mentioned in the keywords but not in the abstract (or in the paper).

Introduction:

Ln. 45-46: poor definition, rephrase.

Ln. 48-50: Odd sentence, consider rephrasing.

Ln. 71-72: Odd, not clear what the authors mean.

Ln. 74: this is anthropomorphic. Judgement bias has been well investigated in a wide range of species who cannot self-report. Rephrase to reflect more relevant literature for the present study.

Ln. 125-128: Vague, does not provide any valuable information. In essence this is blue sky research. If the authors want, see an applied angle (welfare/human animal relationships) they need to make explicit how this would work. Just knowing whether animals have a judgement bias does not improve their welfare!

Ln. 134: Please provide mean age +/- standard deviation or error.

Methods:

Ln. 157-158: It would be good to explain the habituation-dishabituation procedure before the figure.

Ln. 171-189: Please explain what the habituation phase is for? What is the aim?

Ln. 286-292: Could this be moved to the online supplementary materials?

Results:

Ln. 353-354: Should this be moved into the methods section?

Discussion:

Ln. 411: It is not clear how this is linked to welfare at all. Initially these aspects are important to better understand behaviour and cognition of animals. This can be linked to welfare, for example when different individuals potentially perceive the same situation as potentially positive or negative. But all of this needs to be more explicitly elaborated on for the reader.

Ln. 415: Why irrational?

Ln. 456: Do not refer to figures, tables and results of statistical analysis in discussion.

Ln. 490: This has not been tested in the present experiment.

Author Response

Reviewer 2

The topic of the study is interesting. The abstract is insufficient, it is not clear what the study is about and what the key findings are. Too many aspects are mentioned in insufficient detail.

Response: We thank the reviewer for his/her kind comments. We applied each of his/her suggestions and provided the information requested.

The introduction needs a better link between the related topics investigated. The topic of judgement bias is introduced fairly well, personality research is briefly mentioned, but the link is not clear. Why should judgement bias be affected by personality? This needs to be made explicit based on theory. In the abstract, sex and age differences are mentioned. It needs to be explained how, based on theory, judgement bias is expected to differ depending on sex and age.

Response: We added the information suggested.

More details about the experimental procedure are necessary.

How long have the experiments taken? During the pretest habituation phase, donkeys were brought into the testing area for 30 mins a day and the testing period was 20 mins or less depending on if the donkey accomplished the task. In the Pretest- detour phase donkeys had 3 mins to find the bucket. For the judgement bias trial the donkey had 2 mins. Hence, maximum total length of the experiment per animal was 55 min.

In which order have the individuals been tested? Trials were presented in a pseudorandomized order with no more than 2 consecutive trials of the same type. In the judgement bias trial the order was always randomized.

How many tests per day were conducted? A minimum of 3 tests for pretesting and no more than 2 consecutive trails of the same type (loaded vs unloaded bucket) for the Pretest training- detour phase, and for testing the judgement bias, donkeys would complete/attempt 2 sets of buckets in different places, 3 places, so a total of 6 trails.

How much break between the tests? 2 mins per trail for pretest training- detour phase, 3 weeks of training with individual donkey training 1 a day with 12 hours between each trail.

Please provide an overview over number of trials per individual. 3 trials/donkey for pretesting and then during the judgement bias trail, one individual donkey would be presented 3 options twice, total of 6. Order was always randomized in the judgement bias trail.

Have all individuals completed each stage of the experiment before moving on or have individuals been moved through experimental stages at different times? Some donkeys advanced quicker than others, so individuals moved through the experiment at different times for pre-testing and then all individuals were tested for the judgement bias at the same time.

Have all individuals who started the experiment completed it? Yes, all individuals completed the judgement bias testing (Figure 4).

How long did it take each individual to complete each stage of the experiment?   Donkeys were given 20 minutes for the pretest habituation phase and if they had not located the new objects, test was terminated.

Details of analysis of judgement bias are missing. How was the data analysed?

Response: Information was provided.

The authors provide a very detailed analysis on personality clustering, effects of sex, age, weather. All these questions are not properly introduced, and the results section becomes quite confusing. Are these analyses relevant? What does this tell the reader? This is not a major issue and there are two potential fixes: (A) the manuscript could focus on the major research questions and all additional information and analysis can be moved into the supplementary materials; (B) all aspects of the analysis are properly introduced and explained to the reader throughout the manuscript.

Response: We moved the information and analysis to supplementary material as suggested by the reviewer. Sections were reorganized to present results with more coherence.

Presently the presentation of the results can be improved. The authors need to go back to the main research question and provide 1-2 figures illustrating the results answering this research question. I have to admit, after reading the manuscript I am a bit confused what the actual research question is. Is it about judgement bias in donkey, about personality, about sex or age or weather effects? This needs to be made explicit.

Response: Research aims were clarified. The aim of this study was to determine the existence of evidences of interindividual variability in Miniature donkeys in their judgement bias upon an ambiguous stimulus, whether biases may ascribe to personality factors or whether other factors such as sex, age or weather elements may be involved.

Not all factors need to be investigated but only the once for which clear predications can be derived from theory. There needs to be a clear quantitative analysis of personality and the personality aspect needs to be better linked to the judgement bias aspect.

Response: We understand the reviewer suggestion. However, by specifying that we run a Spearman’s rank correlation analysis we inferred the quantification of the relationships that are detected.

As it is generally known, a Spearman correlation of 1 results when the two variables being compared are monotonically related, even if their relationship is not linear. This means that all data points with greater x values than that of a given data point will have greater y values as well.

Are personality differences in judgement bias to be expected?

Response: Significant Spearman’s correlations were detected for some personality features while these were not evidenced for others, hence, personality differences are to be expected. We clarified this in the body text.

Which differences, in which directions? Which personality aspects are expected to be related to an optimistic bias, which aspects are expected to be related to a pessimistic bias? Can this be confirmed by the present study? The results section is too descriptive and too little explorative. You need to tell the reader whether your results support your predictions or not, you cannot simply describe everything and leave it to the reader to figure it out.

Response: This information was present but has been clarified and organized to be more understandable.

Analysis: Could a multivariate approach be taken into consideration? As authors analyse factors like sex, age, weather, it seems appropriate to consider them in the statistical analysis. It seems inappropriate to analyse the effect of each of these factors independently and assessing the contribution to variance in the dataset of all the factors together seems appropriate.

Response: Multivariate approaches are not indicated in this case, for the number and type of data available. Indeed, a correlation analysis was chosen to fit data properties. In our study, data was limited due to the complexity and time needed to perform all the stages in this study, which make it difficult for it to be performed at a large scale. The consideration of personality as a multilevel factor involves that the complexity of the factor combinations that occurs when additional factors are added in a multivariate analysis increases in a way that sample needs to be consistently larger for the study not to be biased by variance explanatory power inflation. Furthermore, multivariate analysis may involve not only testing factors individually at the same time but also their conjoined interactions between these factors, which makes the analysis even more complex. In this context, as suggested in previous research Navas et al (2016.), in behavioral studies, where multiple factors are involved, the reliance on several factors individually, may help us quantify the factors or effects involved more accurately than their conjoint effects. For these reasons, the use of Spearman’s correlations was chosen as to fit data properties.

Navas, Francisco Javier, et al. "Measuring and modeling for the assessment of the genetic background behind cognitive processes in donkeys." Research in veterinary science 113 (2017): 105-114.

Abstract:

Ln. 29: please mention who completed the personality questionnaire. Owners, caretakes, researchers? Please mention which experimental paradigm was used to test judgement bias.

Response: Information requested was added.

Ln. 33: ‘No significant differences were found between sexes (P>0.05).’: Without information on sample size and test used the p value is insufficient information. Either delete p value or provide additional information which would allow the reader to properly judge the information provided.

Response. Information requested was added.

Ln. 33: ‘Vigilance behaviour significantly decreases as age progresses.’ Odd phrasing. I assume this is not a longitudinal study which followed individual donkeys across ages. Rephrase ‘vigilance behaviour significantly decreased with age’. Please put vigilance into context, why was the behaviour relevant, what do changes tell us?

Response: Sentence was rephrased as suggested.

Ln. 33-34: Which results provide this information? Please explain.

Response: Explained.

Ln. 35-36: How is patience related to this? Patience is a different cognitive concept compared to judgement bias. The authors do not explain which experiment they conducted and what they measured. More information needs to be provided in the abstract.

Response: We added the information requested by the reviewer. However, we would like him/her to understand that 200 words do not leave much space to clarify everything.

Ln. 38-41: Over-generalisation of results. Please provide a critical discussion on what the results of the experiments tell the reader (judgement bias, personality).

Laterality is mentioned in the keywords but not in the abstract (or in the paper).

Response: We agree, information was removed.

Introduction:

Ln. 45-46: poor definition, rephrase.

Response: definition was rephrased.

Ln. 48-50: Odd sentence, consider rephrasing.

Response: We rephrased it.

Ln. 71-72: Odd, not clear what the authors mean.

Response: Sentence was removed to make the section more understandable.

Ln. 74: this is anthropomorphic. Judgement bias has been well investigated in a wide range of species who cannot self-report. Rephrase to reflect more relevant literature for the present study.

Response: Sentence was rewritten and literature was changed to correct what the reviewer suggested.

Ln. 125-128: Vague, does not provide any valuable information. In essence this is blue sky research. If the authors want, see an applied angle (welfare/human animal relationships) they need to make explicit how this would work. Just knowing whether animals have a judgement bias does not improve their welfare!

Response: We rewrote the sentence to include what the reviewer suggested.

Ln. 134: Please provide mean age +/- standard deviation or error.

Response: We added mean age and sd.

Methods:

Ln. 157-158: It would be good to explain the habituation-dishabituation procedure before the figure.

Response: Figure was moved after test explanations as suggested by the reviewer. Sections were rearranged.

Ln. 171-189: Please explain what the habituation phase is for? What is the aim?

Response: Added.

Ln. 286-292: Could this be moved to the online supplementary materials?

Response: We think this table must be kept here as one of the main aims is to determine the existence of links between personality features and judgment bias. Otherwise, we moved the personality feature correlation graph to supplementary as suggested in a previous comment.

Results:

Ln. 353-354: Should this be moved into the methods section?

Response: These are the results of the ICC tests which had been presented at the M&M section, hence we think they are appropriately placed.

Discussion:

Ln. 411: It is not clear how this is linked to welfare at all. Initially these aspects are important to better understand behaviour and cognition of animals. This can be linked to welfare, for example when different individuals potentially perceive the same situation as potentially positive or negative. But all of this needs to be more explicitly elaborated on for the reader.

Response: We added more information to approach the reviewer suggestion.

Ln. 415: Why irrational?

Response: We agree. We changed it to unexplored.

Ln. 456: Do not refer to figures, tables and results of statistical analysis in discussion.

Response: We removed it.

Ln. 490: This has not been tested in the present experiment.

Response: We removed it.

Reviewer 3 Report

Summary

The authors evaluated the influence of individual differences, subject characteristics (sex and age), and other environmental aspects (weather and daytime) on the judgment bias task in Miniature Donkeys. The results indicate increasing latency in the task for higher level of patience. Moreover, a negative association between age and vigilance was reported. 

The aim of the study is interesting and relevant. However, several methodological issues prevent me from recommending its publication.

First I will describe the main general critical aspects of the study suggesting possible directions to improve the article. Subsequently, I will list minor issues.

General Issues:

1) The principal critical aspects is the sample size. Having only 12 Donkeys, it is difficult to evaluate individual differences or to conduct any reliable statistical inference. I recognise that working with animals is very demanding and a large sample size could be difficult (or even impossible) to collect. Thus, I do not question the sample size per se but rather the statistical approach and overall interpretation of the results. 

In this scenario (small sample size/ low power), there is huge variability in the estimates. Therefore results are not reliable and, likely, they will not be replicated in future studies. Moreover, filtering for statistical significance in underpowered studies is problematic: only large effects would result "statistically significant" whereas small/medium effects would never be statistically significant. This has two problematic aspects: ignoring small/medium effect that may actually be important, considering large effects that are actually an overestimation of the true effect. In fact, statistically significant effects in underpowered studies are very likely just an overestimation of the true effect.

Regarding this point consider the literature about Design Analysis and Type-M error, for example: Gelman, A., & Carlin, J. (2014). Beyond Power Calculations: Assessing Type S (Sign) and Type M (Magnitude) Errors. Perspectives on psychological science. https://doi.org/10.1177/1745691614551642. 

More generally, consider the following article regarding statistical analysis issues in the case of small sample size in studies of animal behaviour: 
Garamszegi, L. Z. (2016). A simple statistical guide for the analysis of behaviour when data are constrained due to practical or ethical reasons. Animal Behaviour. https://doi.org/10.1016/j.anbehav.2015.11.009

2) A second critical aspect is the data analysis approach and in particular the number of relations evaluated. In the study, the judgement bias task was evaluated according to 18 different personality characteristics, 4 environmental factors, and 3 participants characteristics. Moreover, the authors reported a negative association between age and vigilance. This means that, although "This study aimed to determine repercussions of environmental (weather elements) or subject-inherent factors (sex, age or personality features) on judgment bias." [from the abstract], probably also all relations between personality characteristics and age (maybe also sex) have been evaluated. This leads to an enormous number of tests (there are more tests than subjects), with consequent inflation of the Type I error. Thus, results are not reliable and they should be interpreted with caution. Remind the famous sentence from Ronald Coase: "If you torture the data long enough, it will confess to anything". 

In addition to these analyses, the authors evaluated also all the relation between all personality characteristics and they run a cluster analysis. Having only 12 subjects and 18 personality characteristics this is really questionable. Estimated correlation are subjected to a huge variability and any kind of clustering should be avoided as results would be unreliable.

3) Given the points discussed above (small sample size, high number of relations evaluated), it is clear that this is an exploratory study and any result should be interpreted with caution recognizing its limits. However, the study is very valuable as it provides important indications for future study. I would suggest authors to revise the statistical analysis, clarifying the approach and underling the explorative/descriptive nature of the study. In particular:

- evaluate results reliability in a Leave-One-Out sensitivity analysis. That is, run the analysis excluding one subject at a time to evaluate the variability of the results. Although, Spearman's correlation is robust, having only 12 subjects is a "limit case". I did a trial considering the values in Figure 4 (why there are only 11 points?), excluding one subject at a time, I obtained correlation values ranging from 0.63 to 0.84. This is a lot of variability and should highlight the importance of interpreting the results with caution.

- warn about statistical significance. I would suggest to do not filter for statistical significance but rather focus on descriptive statistics, effect size estimation and confidence intervals. As previously discussed in point 1, a small/medium effect that may be important would never be "significant", whereas "significant" effects will have very large values that are likely an overestimation of the true effect. I know that publishing without p-values is difficult but this point should be at least discussed.

- do not run any cluster analysis. If you consider the variability in the correlation estimates, you realise that any attempt to run a cluster analysis very questionable. Results are not reliable. You could still evaluate correlation among personality dimensions (Table 2) but be aware of the variability of the results.

- since everything that could be tested has been already tested, it does not make sense to restrict the number of relations evaluated. It is just important to clearly report all relations considered and how variables were defined. For example, it is not clear to me if Adj.L scores were computed considering as latency to ambiguous stimulus only An condition or also A- and A+ (if not why they were not considered? Maybe I am just not familiar with the experimental design).

4) Other Issues concern the clarity and logical flow of the article. Several parts are not well connected making the reading rather difficult. Moreover, there are several errors (for example in reporting statistical results, table referencing, grammar errors). Most evident are reported below. I suggest authors carefully revise the article and improve the overall clarity and logical flow of the article. 

Minor Issue:

1) Line 20 - "Individuals were scored on eleven personality traits" should be 18?

2) Line 39 - "vigilance decline is may not be parallel"  remove is?

3) Line 157 - "Nor only personality features may constitute a source of judgment bias variability." It seems something is missing

4) Line 211 - \mu to indicate the mean. Greeks letters are used for population parameters, not sample values. (Recurring error also in formula 1)

5) Line 352 - "From The"

6) Line 475 - missing ")"

7) Line 484 - "85±70.3 seconds (A)" should be An?

8) Figure 4 - Why there are 11 points and not 12? Why did you plot the Pearson-regression line if actually rank correlation was conducted?

9) Line 528 - "(Table 1)" should be Table 3? Also in the table caption

Round 2

Reviewer 2 Report

The entire manuscript should be carefully proof-read. I feel the revision has at times worsen the quality of the manuscript. The writing is unclear at times. The correlation between personality parameters and latency scores (figure 4) is not very convincing. Please provide details of the Spearman correlations (n, r2, exact p values). The authors discuss so many things (personality, age, sex, weather) and the rationale is not entirely clear. Sample size is small (12 individuals) to test a wide variety of factors. Interrelationship between factors, e.g. sex and age, sex and weather is not clear. 

Ln. 38: ‚Sexes did not significantly differ’: odd working, consider revising; what exactly was not different between male and female individuals?

Lns. 39-40: ‘The significant age-dependent progressive vigilance decline is may not be parallel to increasingly poorer performances at rather attentionally complex cognitive tasks given the lack of significant interfeature correlations.’ Complicated sentence, really difficult to understand. Also the relevance of vigilance is not clear, isn’t the experiment about approach latency?

Lns. 41-45: odd sentence, consider rephrasing. Please explain what the ‘latency differences’ are, i.e. are donkeys approaching positive or negative stimulus quicker? Please explain individual differences you describe, e.g. give range of latencies.

Ln. 45: How was ‘patience’ measured?

Lns. 91-92: ‘It is in the framework of the emotions in the background of individuals, in which affect occurs.’ Odd, what do you mean by ‘background of individuals’?

Lns. 131-133: Odd phrasing consider revising.

Ln. 157: Not only.

Ln. 161: What are intersex differences? Differences in intersex individuals or do you mean differences between the sexes?

Ln. 210: Have donkeys been sterilised or castrated?

Ln. 249: trial not trail.

Ln. 254: mins or minutes but be consistent throughout.

Ln. 355: Why calling it ‘Habituation/Training/Testing Area’ wouldn’t it be easier to simply call it ‘experimental area’ or something similar?

Lns. 489-491 and throughout manuscript: Why are some words in capital letters?

Author Response

All the team responsible for this paper acknowledge the comments from the reviewers and editor, as they help to improve the quality of our manuscript. In the following paragraphs, we will describe and address how referees’ new recommendations were followed. A point-by-point response to comments is provided as well as a file where changes are highlighted.

Reviewer 2

Comments and Suggestions for Authors

The entire manuscript should be carefully proof-read. I feel the revision has at times worsen the quality of the manuscript. The writing is unclear at times. The correlation between personality parameters and latency scores (figure 4) is not very convincing. Please provide details of the Spearman correlations (n, r2, exact p values). The authors discuss so many things (personality, age, sex, weather) and the rationale is not entirely clear. Sample size is small (12 individuals) to test a wide variety of factors. Interrelationship between factors, e.g. sex and age, sex and weather is not clear. 

Response: We thank the reviewer for his/her comments as they make our paper improve. We are sorry for the grammar problems and sentence formation. There must have been a problem when using the tracking changes filter which distorted text somehow. The whole manuscript was checked to improve the manner in which information is reported and to remove and correct typos. Sample details for Spearman correlations were provided as requested. However, Spearman's correlation applies to ranks and so provides a measure of a monotonic relationship between two continuous random variables. It is also useful with ordinal data and is robust to outliers (unlike Pearson's correlation). If one uses Pearson's, one could describe the strength of the correlation in terms of shared variance (coefficient of determination, R2) but this cannot be done in Spearman’s correlation. Indeed, it is Pearson’s correlation coefficient which appears in the formula for R2 calculations. This occurs as Spearman's rho, for example, represents the degree of correlation of the data after data has been converted to ranks. Thus, it already captures the strength of relationship. We do not understand what additional information may be provided by exact p values. It would only increase the complexity of the table, which we were suggested to simplify at a previous round, and it would not bring any additional information. Sample size limitations were the reason for us testing effects individually. If we combine or interact factors, statistically, levels within each interaction increase, without having enough individuals to support such subdivision. That is the reason why the statistical approach used in this study (Non-parametric correlations between individual factors) was the most suitable. We aimed at identifying potential conditioning factors for judgement bias, and this was done on the context of the sample that we had.

Ln. 38: ‚Sexes did not significantly differ’: odd working, consider revising; what exactly was not different between male and female individuals?

Response: We agree. We corrected it.

Lns. 39-40: ‘The significant age-dependent progressive vigilance decline is may not be parallel to increasingly poorer performances at rather attentionally complex cognitive tasks given the lack of significant interfeature correlations.’ Complicated sentence, really difficult to understand. Also, the relevance of vigilance is not clear, isn’t the experiment about approach latency?

Response: We agree, we removed this information from the article.

Lns. 41-45: odd sentence, consider rephrasing. Please explain what the ‘latency differences’ are, i.e. are donkeys approaching positive or negative stimulus quicker? Please explain individual differences you describe, e.g. give range of latencies.

Response: We added what the reviewer suggested. Still would like the reviewer to appreciate that only 200 words can be used for the abstract.

Ln. 45: How was ‘patience’ measured?

Response: As described in the body text, the references described in [Navas, et al. [1, 2, 3] were adapted to measure personality traits. In the case of patience, the scale used to measure patience was adapted from the perseverance trait described in the paper.

  1. Navas, F.J., et al., Measuring and modeling for the assessment of the genetic background behind cognitive processes in donkeys. Research in Veterinary Science, 2017. 113: p. 105-114.
  2. Navas González, F.J., et al., Dumb or smart asses? Donkey's (Equus asinus) cognitive capabilities share the heritability and variation patterns of human's (Homo sapiens) cognitive capabilities. Journal of Veterinary Behavior, 2019. 33: p. 63-74.
  3. Navas González, F.J., et al., Nonparametric analysis of noncognitive determinants of response type, intensity, mood, and learning in donkeys (Equus asinus). Journal of Veterinary Behavior, 2020. 40: p. 21-35.

Lns. 91-92: ‘It is in the framework of the emotions in the background of individuals, in which affect occurs.’ Odd, what do you mean by ‘background of individuals’?

Response: We referred to the emotional background of individuals. Citation was added.

Lns. 131-133: Odd phrasing consider revising.

Response: Sentence was rewritten.

Ln. 157: Not only.

Response: Corrected

.

Ln. 161: What are intersex differences? Differences in intersex individuals or do you mean differences between the sexes?

Response: Agree. We corrected it.

Ln. 210: Have donkeys been sterilised or castrated?

Response: Yes, 3 males were castrated, this info was added.

Ln. 249: trial not trail.

Response: Corrected.

Ln. 254: mins or minutes but be consistent throughout.

Response: We used mins consistently throughout the manuscript.

Ln. 355: Why calling it ‘Habituation/Training/Testing Area’ wouldn’t it be easier to simply call it ‘experimental area’ or something similar?

Response. Suggestion was followed and applied across the body text.

Lns. 489-491 and throughout manuscript: Why are some words in capital letters?

Response: The whole manuscript was checked for incorrect capitalization.